# Cellular Senescence and Vitamin D Deficiency Play a Role in the Pathogenesis of Obesity-Associated Subclinical Atherosclerosis: Study of the Potential Protective Role of Vitamin D Supplementation

**DOI:** 10.3390/cells10040920

**Published:** 2021-04-16

**Authors:** Abdulhadi I. Bima, Abdullah S. Mahdi, Fayza F. Al Fayez, Taghreed M. Khawaja, Salwa M. Abo El-Khair, Ayman Z. Elsamanoudy

**Affiliations:** 1Department of Clinical Biochemistry, Faculty of Medicine, King Abdulaziz University, Jeddah 21465, Saudi Arabia; abima@kau.edu.sa (A.I.B.); absmahdi91@gmail.com (A.S.M.); falfaiz@kau.edu.sa (F.F.A.F.); tagreed2009-@hotmail.com (T.M.K.); 2Medical Biochemistry and Molecular Biology, Faculty of Medicine, Mansoura University, Mansoura 35516, Egypt; salwakhair@mans.edu.eg

**Keywords:** obesity, vitamin D, senescence, atherosclerosis, endothelial dysfunction

## Abstract

The exact link between obesity, vitamin D deficiency, and their relation to cellular senescence in the pathogenesis of subclinical atherosclerosis is still under debate. Therefore, the current study aims to verify the possible role of vitamin D deficiency and cellular senescence in the pathogenesis of obesity-related subclinical atherosclerosis. Moreover, it aims to investigate the possible protective role of vitamin D supplementation. Fifty-seven male albino rats were enrolled in the study and classified into four groups: negative (10) and positive control groups (10), an obese model group (24), and a vitamin-D-supplemented obese group (13). Aortic tissue samples and fasting blood samples were collected. The following biochemical investigations were performed: serum cholesterol, triglycerides, HDL-C, LDL-C, ALT, AST, CPK, CK-MB, and hs-cTnt. HOMA-IR was calculated. Moreover, serum SMP-30, 25 (OH)Vitamin D_3_, and eNOS were determined by the ELISA technique. Aortic gene expression of eNOS, SMP-30, and P53 was estimated by real-time qRT-PCR. Serum 25(OH) D_3_ and SMP-30 were lower in the obese group. In addition, the obese group showed higher serum lipid profile, HOMA-IR, eNOS, ALT, AST, CPK, CK-MB, and hs-cTnt than the control groups, while decreased levels were found in the vitamin-D-treated obese group. Gene expression of eNOS and SMP-30 were in accordance with their serum levels. A positive correlation was found between vitamin D level and SMP-30. In conclusion, obesity is associated with vitamin D deficiency and enhanced cellular senescence. They could play a role in the pathogenesis of obesity-associated subclinical atherosclerosis and endothelial dysfunction. Vitamin D supplements could play a protective role against such obesity-related comorbidity.

## 1. Introduction

Obesity is defined as abnormal extensive fat accumulation that negatively affects health [1]. It is defined as a body mass index (BMI) ≥ 30 kg/m^2^ [2]. The increasing prevalence of obesity is a worldwide health concern because excess weight gain causes an increased risk for several diseases, such as type 2 diabetes, hypertension, stroke, heart disease, cancer, gallbladder disease, depression, osteoarthritis, metabolic syndrome, and sleep apnea. Moreover, dyslipidemia and delayed wound healing are recorded as obesity-related problems [3].

It is reported that obesity is associated with low serum 25 Hydroxycholecalciferol [25(OH)D_3_] [4]. In addition to the proven bone-related health hazards of vitamin D deficiency, other possible consequences of vitamin D deficiency are reported and related to heart diseases such as clinical and subclinical atherosclerosis, hypertension, diabetes, and even stroke [5].

Cellular senescence is defined as the deterioration of the cells as a result of aging [6]. It is an essentially irreversible growth arrest of the cell that occurs in response to various cellular stressors, such as telomere erosion, DNA damage, oxidative stress, and oncogenic activation [7]. Cellular senescence plays a role in the development and progression of numerous diseases, such as obesity, while on the other hand, obesity could be a risk factor for accelerating the rate of cellular senescence [8].

Atherosclerosis is a type of prolonged vessel inflammation disease that exhibits a long asymptomatic phase. The progression of this vascular abnormality could ultimately result in more severe cardiac-related complications [9]. Obesity is thought to affect cardiovascular events. Indeed, the concept of “metabolically healthy” obesity has been studied, and it has been shown that coronary heart disease may be increased in this obesity phenotype [10].

Low 25(OH)D_3_ concentrations increase the risk of hypertension, peripheral vascular disease, diabetes mellitus, myocardial infarction, heart failure, and cardiac mortality [11]. Moreover, it is reported that low 25(OH)D levels are associated with endothelial dysfunction, inflammation, increased vascular stiffness, and high coronary artery calcium scores [12]. Moreover, it is reported that shortened telomere as an early sign and characteristic feature of cellular senescence is a prognostic biomarker for the early identification of subjects at high risk of developing CVD before symptoms appear [13].

The exact link between obesity, vitamin D deficiency, and their relation to cellular senescence in the pathogenesis of subclinical atherosclerosis is still under debate. Therefore, the current study aims to verify the previously reported link between vitamin D status and obesity, to study their possible contribution to cellular senescence, and finally to investigate the possible protective role of vitamin D supplementation against obesity-related subclinical atherosclerosis.

## 2. Methods

This study is an experimental animal study, conducted in the Department of Clinical Biochemistry-Faculty of Medicine, King Abdulaziz University, Jeddah, Saudi Arabia.

### 2.1. Animals and Experimental Protocol

Sixty-seven adult male Sprague–Dawley albino rats, weighing approximately 120 ± 20 g, were obtained from the animal house, Faculty of Pharmacy, King Abdulaziz University, and overall, 57 rats were enrolled in this study after the death of three rats (during the 3 months of the experiment) and the failure of seven rats to gain weight. Animals were housed (at 24 °C ± 3 °C) in a temperature-controlled room with 40–70% humidity and a 12/12 h light/dark cycle in metal cages with meshes, and they were continued on commercial food, ad libitum, consisting of standard laboratory rat chow and free access to drinking water for the first 10 days following delivery to allow acclimation to the new environment. The experiment was performed in accordance with the animal welfare act and the guide for care of the Experimental Research Center, King Abdulaziz University.

The rats were classified into 4 groups as follows. Group I served as a negative control group and included 10 albino rats with a normal diet. They were given water ad libitum and were fed a standard chow with 26.5% protein, 3.8% fat, 40% carbohydrate, and 4.5% crude fiber in 100 g of chow [14]. Group II served as a positive control group and included 10 albino rats fed a normal diet containing 500 IU/kg vitamin D supplement. This started from the 3rd week and continued for three months’ duration, according to the method described by Farhangi et al. [15]. Group III served as an obese animal group (non-treated obese group model) and included 24 albino rats. They were fed, ad libitum for 3 months, a high-carbohydrate diet with 20% protein, 15% fat, 60% carbohydrate, and 2.2% dietary fiber in 100 g of chow [14]. Group VI served as an obese animal group (vitamin D-treated obese group) and included 13 albino rats fed as in group III [14] + vitamin D supplement with a dose of 500 IU/kg/day, starting from 3rd week and continuing for three months’ duration [15].

Female or non-albino rats, rats weighing more than 200 g at the start of the research experiment, and rats that failed to gain the expected weight (7 rats) were excluded from the study.

### 2.2. Anthropometric Measurement

Weekly measurements of height in cm and body weight in grams were carried out, and body mass index (BMI) was calculated according to Novelli et al. [14].

### 2.3. Sampling and Biochemical Investigations

At the end of the experimental protocol duration, after 12 h of overnight fasting, each rat was anesthetized by ether, and a blood sample was withdrawn under a complete aseptic condition from retro-orbital venous plexus using a disposable plastic syringe. The blood samples were collected into plain plastic containers, and sera were collected after centrifugation at 15,000 rpm for 20 min, then divided into aliquots, and stored at −80 °C until the biochemical investigations were performed.

All rats from each group (total 57) were sacrificed under diethyl ether anesthesia, and the aorta was removed for molecular studies. The aorta was excised, weighed, divided, and immediately frozen in liquid nitrogen. The liquid-nitrogen-frozen aortic tissue sample (30–50 mg in weight) was used for total RNA extraction and real-time qRT-PCR analysis of expression of eNOS, SMP-30, and P53.

Serum total cholesterol (CHOL), triglycerides (TG), HDL-C, and LDL-C were measured colorimetrically using kits provided by Abbott Diagnostics. Activities of alanine transaminase enzyme (ALT) and aspartate transaminase enzyme (AST) were estimated by enzymatic methods (Sigma-Aldrich, St. Louis, MO, USA). CPK and CK-MB were determined by spectrophotometer using the commercially available kits (the instrument used was the Architect c8000 and the kit was provided by Abbott Diagnostics company). High sensitivity cardiac Troponin T (hs-cTnt) was determined by chemiluminescent microparticles immunoassay technology by Architect i2000; the kit was provided by Abbott Diagnostics. Fasting blood glucose was estimated by a colorimetric method (AGAPPE Diagnostics Switzerland GmbH, Cham, Switzerland) using a spectrophotometer.

Moreover, serum levels of SMP-30, 25 Hydroxy-vitamin D3, insulin, and eNOS were determined by using enzyme-linked immunosorbent assay (ELISA) technique. eNOS was measured using a rat endothelial nitric oxide synthase (eNOS) ELISA kit (competitive ELISA, catalog number MBS721860, MyBioSource, San Diego, CA, USA). Serum vitamin D level was measured using rat 25-hydroxy vitamin D_3_ (25 (OH) D_3_) ELISA kit (catalog number MBS261766, MyBioSource, Diego, CA, USA). SMP-30 in rat serum was measured using Rat RGN/Regucalcin ELISA kit (Sandwich ELISA)-LS-F38749 (LifeSpan Biosciences, Seattle, WA, USA). The serum level of insulin was estimated using Ray-Bio rat insulin ELISA kit, an in vitro ELISA kit for the quantitative measurement of rat insulin in serum and plasma.

In each sample, the degree of insulin resistance was estimated by the homeostasis model assessment of insulin resistance (HOMA-IR) as described by Matthews et al. [16]. HOMA-IR was calculated by taking into account fasting insulin and fasting blood glucose levels according to the equation: fasting insulin (μU/mL) × fasting plasma glucose (mg/dL) × 0.0551/22.5

### 2.4. Molecular Study of Gene Expression of eNOS, SMP-30, and P53

Real-Time qRT-PCR: Total RNA was isolated from 30–50 mg tissue samples of rats’ aorta after snap freeze with liquid nitrogen using Tri-Fast TM reagent (PeqLab, Biotechnologie GmbH, Carl-Thiersch St. 2B 91052, Erlangen, Germany, cat. no. 30-2010), triazole, and chloroform, according to the manufacturer’s instructions. Extracted RNA concentration and purity were determined by NanoDrop™ 2000 spectrophotometer (Thermo Scientific, Waltham, MA, USA).

Reverse transcription reaction for cDNA synthesis was performed with ≈250 ng total RNA using Maxima First Strand cDNA Synthesis Kit (Thermo Scientific, Waltham, MA, USA, cat. no. #K1641). The rat blood mRNA expression of eNOS, SMP-30, and P53 were quantified by real-time PCR on the Applied Biosystem 7500, real-time PCR detection system (Life Technology, Carlsbad, CA, USA) with Applied Biosystem SYBR^®^ Green PCR Master Mix (2X) (Life Technology, USA, cat. No. 4344463). Reaction mixtures were incubated for 10 min at 95 °C, followed by 40 cycles of 15 s at 95 °C, 1 min at 60 °C, and, finally, 15 s at 95 °C, 1 min at 60 °C, and 15 s at 95 °C. The primer sequences used were rat eNOS: forward, 5′-ACCGCCACACAGTAAATCCA-3′; reverse, 5′-TGCCAACAGGAAGCTGAGAG-3′ [17], rat SMP-30: forward, 5′-AGGCATCAAAGTGTCTGCTGTTT-3′; reverse, 5′-GACTGTCGAAGTGCCACTGAACT-3′ [18], rat P53: forward, 5′-CCTATCCGGTCAGTTGTTGGA-3′; reverse 5′-TTGCAGAGTGGAGGAAATGG-3′ [19]. The primers sequences for rat β-actin were 5′-AT-GGTGGGTATGGGTCAG-3′ (forward) and 5′-ATGCCGTGTTCAATGG-3′ (reverse) [18]. The expression of the analyzed genes was normalized to that of the internal control gene, the β-actin, using the comparative ∆∆CT method.

### 2.5. Histological and Morphometric Study

Specimens from the aorta were fixed in Bouin’s solution, paraffin blocks were prepared, and 5 um thick sections were stained with hematoxylin and eosin (H&E). Leica Qwin 500 image analysis software was used to examine the sections on an IBM-operated computer system (Leica Microsystems, Wetzlar, Germany). The thickness of the intima and media was measured, as it is difficult to discriminate between intima and media in aorta as an elastic artery.

### 2.6. Statistical Analysis

The collected data were entered and analyzed using SPSS statistical software (version 20). The results were presented as means (±SD) or as frequencies according to variable types. We compared the different variables according to BMI classes through a one-way ANOVA test and independent *t*-test. We also used the post hoc test (Fisher’s least significant difference (LSD)) for multiple comparisons. Pearson correlation coefficient was used to make correlations between different variables.

## 3. Results

In the current study, 67 male albino rats were enrolled in the experimental procedure at the start of the study, while three rats died during the study period and seven rats failed to gain weight. The remaining 57 rats were classified as negative and positive control groups (10 rats/group), as a model of obesity (24 rats), and as the obese group that was supplemented with vitamin D (13 rats). Comparison of the anthropometric, biochemical data between the studied groups are presented in Table 1. Obese group shows a significant increase in weight and BMI in comparison to the other groups.

The glucose, insulin, and HOMA-IR variables are significantly higher (*p* < 0.001) in the obese group compared with the other group. Regarding the lipid profile of the studied groups, the serum triglyceride, total cholesterol, and LDL-C variables are significantly higher in the obese group, and HDL-C is significantly higher in the positive control group (*p* < 0.001). Moreover, there is no significant change in the level of both calcium and phosphorus level in all studied groups (*p* = 0.61 and 0.72, respectively).

Looking at the data of the biomarkers of endothelial dysfunctions and subclinical atherosclerosis that are presented in Table 1, hs-Tnt, CK, CK-MB, AST, ALT, and eNOS are significantly higher in obese groups than other groups (*p* < 0.001).

Comparison of serum 25(OH)D_3_ and SMP-30 between the studied groups shows that serum 25(OH)D_3_ is significantly higher in the group obese + vitamin D than other groups (*p* < 0.001). Moreover, SMP-30 serum levels show the lowest levels in the obese groups with a significant increase in the obese group that is supplemented with vitamin D.

To investigate levels of biomarkers of endothelial dysfunctions, subclinical atherosclerosis, and associated cellular senescence in blood vessels, aortic mRNA gene expression of eNOS, SMP-30, and P53 were estimated and presented in Table 2 and Figure 1. eNOS gene expression was in accordance with its serum level, which was significantly highly increased in the obese group. This was accompanied by a significant increase in the obese group of P53 gene expression, while the cellular senescence marker SMP-30 was significantly lowered in the obese group, and came back to around its normal level with supplementation of vitamin D.

Correlations of serum level of 25(OH)D_3_ and SMP-30 with all anthropometric and measured parameters are presented in Table 3 and Table 4. There is a significant correlation in all anthropometric parameters with serum 25(OH)D_3_ level, except for height. Serum 25(OH)D_3_ has a significant negative correlation between weight and BMI. Among the four studied rats’ groups, there is no correlation found between serum 25(OH)D_3_ level and other parameters. Serum 25(OH)D_3_ level shows a negative correlation with insulin, CK-MB, CK, LDL-C, TG, AST, ALT, cholesterol, glucose, eNOS, and hs-cTnt. However, it has a positive correlation with HDL-C.

Regarding the negative control group of rats, there is a positive correlation between serum 25(OH)D_3_ level and blood glucose. A positive correlation among the same group is also detected between serum 25(OH)D_3_ level and CK-MB. A negative correlation between serum 25(OH)D_3_ level and AST is found in the positive control group. In the fourth group (obese + vitamin D supplement), there is a positive correlation between serum 25(OH)D_3_ with glucose and cholesterol. For the other studied parameters in the four studied groups, there is no significant correlation found between 25(OH)D_3_ and other parameters.

In Table 4, correlations of serum level of SMP-30 (the marker of cellular senescence) with all anthropometric and measured parameters are presented. There is a significant correlation in all anthropometric parameters with SMP-30 level except for height. SMP-30 level shows a negative correlation with weight and BMI. In the vitamin-D-supplemented obese group, there is a positive correlation between SMP-30 level and weight, while there is no significant correlation between the SMP-30 level and other parameters.

Regarding the correlation between SMP-30 and the measured biochemical parameters, SMP-30 level has a negative correlation with insulin, eNOS, hs-cTnt T, CK, CK-MB, AST, ALT, TG, LDL-C, cholesterol, and glucose. However, it has a positive correlation with HDL-C. There is a negative correlation between SMP-30 level and HDL-C in the negative control group, and a positive correlation between SMP-30 level and CK in the positive control group.

Lastly, there is a significant positive correlation between serum levels of SMP-30 and 25(OH)D_3_. However, among each studied group separately, there is no correlation found between SMP-30 and serum 25(OH)D_3_ level. The results are presented in Table 3.

### Histological Results

Light microscopic examination of haematoxylin- and eosin-stained sections of the wall of the aorta of the control groups (the negative and positive control groups) showed a thin wall formed of the typical three tunicae: tunica intima, tunica media, and tunica adventitia. Tunica intima revealed simple squamous endothelium with underlying lamina propria. Tunica media appeared with concentrically arranged smooth muscles and parallel elastic membranes. Tunica adventitia showed vasa vasorum (Figure 2A).

In the obese group, there was loss of endothelial cells, and a thickening of tunica intima and media with protrusion of intima into the lumen. Apparent thickening of tunica adventitia with the presence of fat and inflammatory cells and congested vasa vasorum were evident (Figure 2B,C). There is a significant increase in the IMT of the wall of the aorta (Table 1).

In the vitamin-D-supplemented obese group, there was a relative improvement of the wall of the aorta, apart from some lost endothelial cells (Figure 2D).

## 4. Discussion

The interplay between obesity, vitamin D deficiency, and cellular senescence in the pathogenesis of subclinical atherosclerosis is still under investigation. Moreover, the possible protective effect of vitamin D supplements is still debatable regarding its role in endothelial dysfunction.

In the present study, there is a decrease in serum 25(OH)D_3_ level in the rat model group in comparison to other groups. This indicates an association between obesity and vitamin D deficiency using the cut-off <50 nmol/L [20,21]. Our result confirms the association between vitamin D insufficiency/deficiency and obesity. This association has been documented, and a high risk of developing vitamin D insufficiency has been demonstrated in obese individuals in Canada [22], Russia [23], and Saudi Arabia [24,25].

In the present study, a negative correlation was observed between serum 25(OH)D_3_ and BMI. This confirms the previously published inverse relationship between vitamin D status and BMI and WC that has been found [26]. The inverse relationship between vitamin D serum levels and BMI could be explained by many hypotheses. The volumetric dilution hypothesis is the first explanation [27,28]. The second explanation is the VD sequestration in the adipose tissue (AT) hypothesis [29]. Impaired hepatic 25-hydroxylation is the third mechanism that might explain our result [30]. Disturbed VD metabolism in adipose tissue is another mechanism that explains the link between VD deficiency and obesity, as downregulation of gene expression in VD-metabolizing enzymes was observed in obese people, which suggests that AT could be involved in the metabolism of VD [29,31]. All mentioned hypotheses are postulated to explain obesity as a possible cause of vitamin D deficiency [32]. The reverse effect is also acceptable, as vitamin D deficiency could be considered as a cause of obesity rather than a consequence. This relationship can be explained as follows: lowered vitamin D level induces an increased parathyroid hormone level, which leads to enhanced calcium inflow in adipocytes and stimulates lipogenesis. This is proved by the fact that the active form of VD, 1,25(OH)D, inhibits adipogenesis through actions modulated by VD receptors. Furthermore, 1,25(OH)D_3_ can maintain the WNT/β-catenin pathway, which is downregulated during adipogenesis, and thus can inhibit adipogenesis [33]. Therefore, lower VD levels could lead to enhanced differentiation of pre-adipocytes to adipocytes and, consequently, to the development of obesity [31].

Decreased serum SMP-30 protein level in the obesity model group is a finding observed in the current study. This indicates that enhanced cellular senescence mechanism is linked to obesity, which coincides with the published data of Burton and Faragher [8]. This could be explained by obesity-associated high-calorie intake inducing oxidative stress in the adipose tissue, and consequently enhancing the senescence pathway. This leads to disturbed senescent markers, such as increased expression of p53, and senescence-associated beta-galactosidase activity (SA-β-Gal) and pro-inflammatory cytokines with decreased expression of SMP-30, like our result. Moreover, p53 gene upregulation in adipose tissue promoted insulin resistance [34]. This link between obesity and enhanced cellular senescence was confirmed in our study at the tissue gene expression level, as indicated from the increase in P53 mRNA and the decrease in SMP-30 mRNA gene expression in aortic tissue.

The positive correlation between vitamin D and SMP-30 in the current study indicates the possible contribution of its deficiency in the enhancement of cellular senescence in the obese animal group. These results are in agreement with Pusceddu et al. [35], Berridge [36], and Carrillo-Vega et al. [37]. Pusceddu et al. [35] explained this association by the proven function of vitamin D. Vitamin D might play a role in telomere biology and genomic stability. This function may be disturbed in an obesity-associated vitamin-D-deficiency state. Berridge [36] stated that vitamin D deficiency is associated with disordered mitochondrial function through disturbed mitochondrial respiratory chain, downregulation of the gene expression of the uncoupling protein (UCP), reduction in the nuclear mRNA molecules and proteins that contribute to mitochondrial respiration [38], and inhibited expression of sirtuin (SIRT) 1, which plays a role in reducing brain aging and contributes to mitochondrial biogenesis by activating PGC-1α [36] (Carrillo-Vega et al. [37] confirmed the interplay between vitamin D deficit, increased cellular senescence, and obesity).

Increased insulin resistance markers and dyslipidemia are observed in the obese animal group of the current study. These markers are negatively correlated with both vitamin D and SMP-30. This indicates the possible involvement of vitamin D deficiency and enhanced cellular senescence in obesity-associated insulin resistance and dyslipidemia. It is proved that VD deficiency is associated with hyperglycemia, hyperinsulinemia, impaired β-cell function, insulin resistance, and increased HOMA-IR, as presented in the current study. These results are proved by a decrease in HOMA-IR after VD treatment in overweight and insulin-resistant individuals. It is also reported that the concentration of intrahepatic lipids strongly correlates with markers of insulin resistance and confirms the development of dyslipidemia in vitamin-D-deficient states [31,39].

The involvement of increased cellular senescence in the development of insulin resistance was recently reported by Chow et al. [40]. This finding could be explained by the fact that cellular senescence leads to a state of permanent growth arrest, triggered by p53 and p16 INK4A tumor suppressor pathways, and consequently, altered glucose metabolism. Moreover, the link between altered carbohydrate metabolism and cellular aging is explained by the burden of senescent cells, accumulated G6P, which results from the action hexokinase–II enzyme has on its glucose substrate, activating MondoA–MLX-mediated Sik2 transcription. SIK2 phosphorylates p35 priming for the ubiquitin ligase PJA2, which is followed by its proteasome-mediated degradation, which leads to insulin resistance (IR), which results from chronic insulin exposure [39].

Looking back to the main objective of the current study, we aimed to investigate the relationship between obesity, subclinical atherosclerosis, and endothelial dysfunction. The obesity model group of rats shows a higher serum level of cardiac enzymes (total CPK, MB-CPK, ALT, and AST), cardiac markers, hs-cTnT, and eNOS as a marker of endothelial dysfunction. These data proved the possible causative role of obesity as a risk factor of atherosclerosis, even if it is subclinical and endothelial dysfunctions in apparently healthy obese rats. The significant increase in eNOS mRNA level at aortic tissue in the obese group provides more confirmation of the link between obesity and endothelial dysfunction and subclinical atherosclerosis at the blood vessel tissue level.

These findings are in agreement with Csige et al. [41] and Sugiura et al. [42]. It is reported that obesity has an association with abnormal function and structure of myocardial muscle with an increased risk for the development of subclinical myocardial damage, as indicated by higher levels of cardiac troponin-T(hs-cTnT) [43], a similar result to the current study. Therefore, obesity has been independently linked to elevated hs-cTnT levels, and the combination of obesity and elevated hs-cTnT is associated with increased risk of myocardial damage [44] through the biochemically proved endothelial dysfunctions and subclinical atherosclerosis.

The significant negative correlation of the measured biochemical parameters (cardiac and endothelial markers) with serum calcidiol and SMP-30 levels indicates a direct association with vitamin D deficiency and the possible involvement of cellular senescence in the pathogenesis of endothelial dysfunction, and consequently, atherosclerosis.

The results of the current study confirm the role of vitamin D deficiency as a possible causative risk factor of subclinical atherosclerosis, as published by Aydin et al. [45]. Like our results, it is reported that low 25(OH)D_3_ levels are associated with endothelial dysfunction, inflammation, increased vascular stiffness, and high coronary artery calcium scores [12]. Additionally, 25(OH)D_3_ deficiency disturbs vascular functions by increasing the incidence of atherosclerosis, while 25(OH)D_3_ supplementation may play a protective role against atherosclerosis pathogenesis [46]. Low 25(OH)D_3_ levels are also associated with atherosclerosis, such as increased carotid intima-media wall thickness at early stages and vascular calcification, which is a major determinant of morbidity of the patients affected with atherosclerosis [33], as proved in the current study.

The involvement of vitamin D deficiency in the pathogenesis of subclinical atherosclerosis could be explained as follows. 25(OH)D_3_ receptors are expressed in vascular smooth muscle cells, macrophages, cardiomyocytes, endothelium, and lymphocytes [47]. Calcidiol stimulates prostacyclin production by vascular smooth muscle cells, which consequently prevents thrombus formation, cell adhesion, and smooth muscle cell proliferation. These vascular smooth muscle cells and endothelial cells also express 25(OH)D_3_ receptors and have the ability to convert calcidiol to its active form, calcitriol [48]. Moreover, it is reported that 25(OH)D_3_ insufficiency was associated with increased arterial stiffness and endothelial dysfunction [49]. Loss of the role of vitamin D in suppressing foam-cell formation by reducing macrophage cholesterol uptake plays an additional role in atherosclerosis development [50]. Finally, the lack of 25(OH)D_3_ anti-inflammatory actions is an additional mechanism, as the inflammatory process plays an important role in the pathogenesis of atherosclerosis initiation, progression, and thrombus formation [47].

Increased cellular senescence as a consequence of vitamin D deficiency could play a role in the pathogenesis of endothelial dysfunction and atherosclerosis. The results of the present study confirm similar results to Katsuumi et al. [51].

Increased cellular senescence, which is an indicator of arterial aging, is characterized by an increase of the intima thickness in relation to media thickness, which leads to gradual loss of vascular smooth muscle cells’ contractile function, increased arterial permeability, and the development of atherosclerotic disease. Moreover, increased collagen production and a corresponding decline of the elastin content, reduced elasticity/distensibility, and increased stiffness, resulting in a higher systolic blood pressure and lower diastolic pressure, are prominent features of senescent vasculature [51]. All of these mechanisms, in addition to the associated dyslipidemia and insulin resistance state, could contribute to linking vascular senescence to the development of endothelial dysfunction and subclinical atherosclerosis in obesity that is complicated with vitamin D deficiency.

The current study shows an improvement of the measured biochemical parameters in a vitamin-D-treated group of rats. Improvement of the glycemic and insulin sensitivity parameters that are observed in our study coincides with the results of Imga et al. [50]. Higher serum levels of 25(OH)D_3_ after treatment are correlated with better β-cell function, and lowered blood glucose level is detected in subjects at risk for type 2 diabetes mellitus (T2DM). The possible mechanisms by which vitamin D supplementation can improve insulin sensitivity are: vitamin D induces insulin receptor gene expression at the transcriptional level, stimulates insulin-dependent glucose oxidation and thus improves insulin sensitivity, and mediates the highest insulin action through enhancement of signal transduction by regulating extracellular calcium [52].

Regarding regression of dyslipidemia in the vitamin-D-treated group in the current study, the current results are in agreement with Imga et al. [52]. After vitamin D supplementation, Imga et al. [52] observed reduction of LDL-C in both overweight and obese subjects and TG reduction only in overweight subjects in their study. A positive correlation between serum 25(OH)D level and apolipoprotein AI was also evident. This could be explained by vitamin D stimulating the synthesis of HDL-C particles and improving serum lipid levels by the effect of 1,25(OH)_2_D_3_ in regulating the process of adipogenesis [52].

SMP-30 expression increased in the vitamin D group, and vitamin D serum level is positively correlated with SMP-30 serum level. This decrease in cellular senescence with vitamin D supplementation was also obvious at the tissue level, as indicated from the increase of SMP-30 gene expression and the decrease in P53 gene expression in aortic tissue in the vitamin D supplementation group. Downregulation of cellular senescence by the effect of vitamin D supplementation was previously reported by Chen et al. [53]. They stated that higher vitamin D levels after supplementation was correlated with longer telomere length, which means a positive association was established between 25(OH)D levels and telomere length, and consequently, delayed cellular senescence. 25(OH)_2_D_3_ also has an antioxidant effect by transcriptional regulation of Nrf2 mediated through VDR. Moreover, it inhibits oxidative stress, DNA damage, cell senescence, and SASP, and stimulates cell proliferation with upregulation of Bmi1 as well as downregulation of p16, p53, and p21 [53].

Cellular senescence and aging is not a single process. It is mediated by many cellular processes such as autophagy, mitochondrial dysfunction, inflammation, oxidative stress, epigenetic changes, DNA disorders, and alterations in Ca^2+^ and reactive oxygen species (ROS) signaling. All of these cellular aging processes are regulated by vitamin D, which proves its role in inhibiting cellular senescence [36].

Vitamin D supplementation inhibits pro-inflammatory states and oxidative stress, so obesity management besides supplementation with vitamin D can help reduce the risks of CVD and inhibit the release of inflammatory mediators, as a marked reduction of endothelial dysfunction and inflammatory biomarkers was observed in obese non-diabetic individuals treated with vitamin D in a study by Iurciuc et al. [54]. The results of the current study confirmed these published data, as there is an improvement of endothelial markers and markers of atherosclerosis. This improvement was at both the blood and tissue level, as indicated by the improvement of the eNOS protein level in blood and the mRNA gene expression level in the aortic tissue. It is reported that active metabolites of vitamin D bind to the VDR, which regulates the genes in essential processes and has a role in pathways of cardiovascular diseases, including inflammation, thrombosis, and the renin-angiotensin system. Moreover, vitamin D induces vasoactive prostacyclin production in vascular smooth muscle cells and consequently plays a protective role in the development of atherosclerosis. Homocysteine, high-sensitivity C-reactive protein (CRP), cystatin-C, uric acid, and HbA1c were also improved by vitamin D therapy [55].

### Limitation of the Study

One of the main important limitations of the current study regards the difficulty of transferring rodent atherosclerosis phenotypes to a human setting, as most rodent models of atherosclerosis do not completely mimic the pathophysiology of the atherosclerotic artery in human. It is reported that atherosclerotic plaques could frequently be manifested in the wall of the aorta as well as the large vessels, and seldom affect the coronary arteries as in humans [56,57]. Consequently, the validity of the rat model of atherosclerosis used in the current study is highly acceptable at the level of large blood vessels, but not at the level of small blood vessels, which needs a different study model for addressing them.

## 5. Conclusions

Obesity is associated with vitamin D deficiency and increased cellular senescence. Endothelial dysfunction and subclinical atherosclerosis are common complications of obesity, even in an asymptomatic state. There is interplay between vitamin D deficiency and increased cellular senescence in the pathogenesis of this obesity-associated subclinical atherosclerosis. Vitamin D supplementation plays a protective role against such obesity-related comorbidity. Further investigations are required to confirm the results of the current study and to be applied clinically in a human-based study.

## Figures and Tables

**Figure 1 cells-10-00920-f001:**
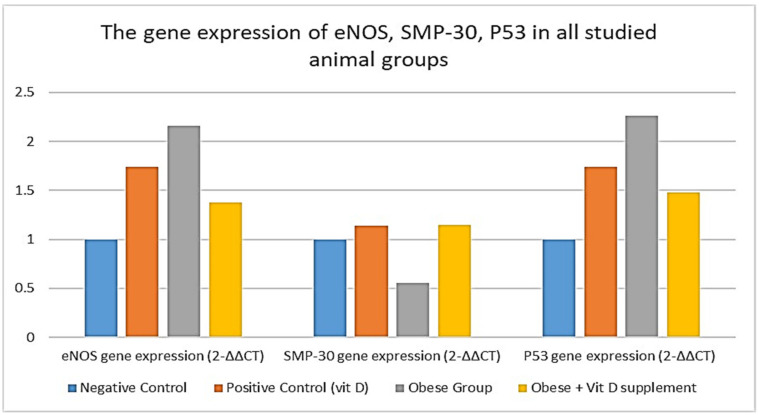
The gene expression of eNOS, SMP-30, and P53 in all studied animal groups.

**Figure 2 cells-10-00920-f002:**
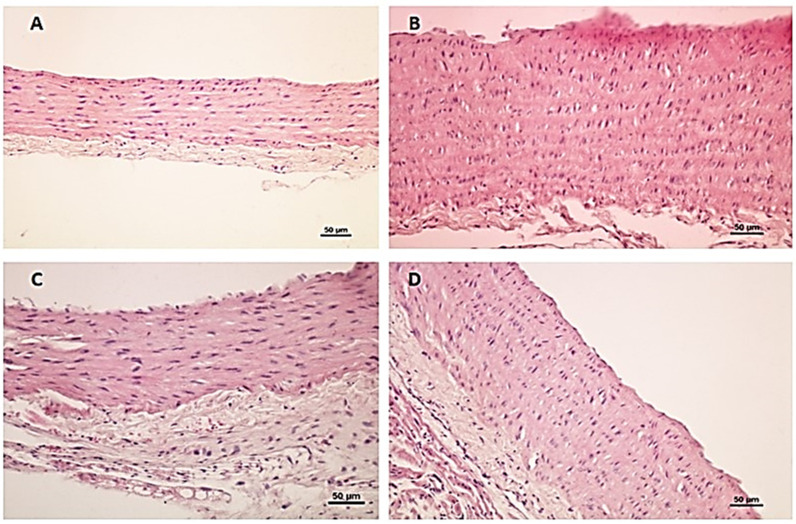
Histological examination of the aorta of the different studied groups. (**A**) A section from the aorta of a control rat showing a thin wall formed of tunica intima, tunica media, and tunica adventitia. Tunica intima reveals simple squamous endothelium with underlying lamina propria. Tunica media is formed of concentrically arranged smooth muscle with parallel internal elastic membranes. Tunica adventitia shows vasa vasorum. (**B**) A section from the aorta of a diseased rat showing loss of endothelial cells, and thickening of tunica intima and media with protrusion of intima into the lumen. (**C**) A section from the aorta of a diseased rat showing apparent thickening of tunica adventitia with presence of fat cells and inflammatory cells and congested vasa vasorum. (**D**) A section from the aorta of treated rats showing the relative improvement of the wall, apart from some lost endothelial cells.

**Table 1 cells-10-00920-t001:** The anthropometric and biochemical parameters of all studied animal groups.

	Negative Control *n* = 10	Positive Control (Vit D) *n* = 10	Obese Group *n* = 24	Obese+ Vitamin-D Supplement *n* = 13	ANOVA *p* Value
Weight (gm)	194.80 ± 5.71	199.70 ± 8.48 ^c,d^	336.33 ± 24.35 ^a,b,d^	308.61 ± 22.39 ^a,b,c^	<0.001 *
Height (cm)	21.26 ± 1.40	21.22 ± 0.83	21.56 ± 1.00	21.38 ± 0.70	0.767
BMI (gm/cm2)	0.43 ± 0.06	0.44 ± 0.03 ^c,d^	0.74 ± 0.07 ^a,b^	0.72 ± 0.07 ^a,b^	<0.001 *
Intima/Media Thickness (IMT) (μm)	59.5 ± 2.9	60.4 ± 3.1	98.6 ± 8.1 ^a,b^	65.3 ± 4.2 ^c^	<0.001 *
Insulin (uIU/mL)	2.21 ± 0.13	1.93 ± 0.07 ^a,c,d^	4.80 ± 0.29 ^a,b,d^	2.30 ± 0.17 ^b,c^	<0.001 *
Glucose (mg/dL)	113.22 ± 16.00	113.45 ± 16.91 ^c^	206.13 ± 117.20 ^a,b,d^	102.30 ± 18.67 ^c^	<0.001 *
HOMA-IR	0.60 ± 0.10	0.55 ± 0.08 ^c^	2.45 ± 1.38 ^a,b,d^	0.58 ± 0.10 ^c^	<0.001 *
TG (mg/dL)	133.11 ± 7.00	118.15 ± 11.27 ^c,d^	269.41 ± 27.05 ^a,b,d^	166.37 ± 6.57 ^a,c,d^	<0.001 *
CHOL (mg/dL)	182.40 ± 4.41	112.87 ± 6.75 ^a,c,d^	268.11 ± 10.79 ^a,b,d^	216.54 ± 17.87 ^a,b,c^	<0.001 *
LDL-C (mg/dL)	130.93 ± 8.84	109.85 ± 6.61 ^a,c,d^	241.07 ± 5.35 ^a,b,d^	153.19 ± 3.70 ^a,b,c^	<0.001 *
HDL-C (mg/dL)	52.99 ± 3.66	81.58 ± 3.79 ^a,c,d^	39.11 ± 1.67 ^a.b.d^	45.18 ± 3.10 ^a,b,c^	<0.001 *
eNOS (ng/mL)	0.37 ± 0.15	0.56 ± 0.25 ^c^	1.54 ± 0.55 ^a,b,d^	0.82 ± 0.65 ^a,c^	<0.001 *
hs-cTnt (pg/mL)	1.72 ± 0.36	2.96 ± 1.83 ^c^	6.11 ± 1.83 ^a,b,d^	2.34 ± 1.45 ^c^	<0.001 *
CK_MB (U/L)	191.66 ± 28.23	92.80 ± 7.62 ^a,c,d^	683.49 ± 140.71 ^a,b,d^	367.84 ± 51.84 ^a,b,c^	<0.001 *
CK (U/L)	499.40 ± 123.60	275.00 ± 56.23 ^c,d^	3365.41 ± 707.24 ^a,b,d^	774.07 ± 74.98 ^b,c^	<0.001 *
ALT (U/L)	40.50 ± 3.62	19.80 ± 5.05 ^a,c,d^	83.87 ± 6.25 ^a,c,d^	56.38 ± 6.18 ^a,b,d^	<0.001 *
AST (U/L)	141.20 ± 13.90	53.10 ± 8.46 ^a,c,d^	250.33 ± 16.00 ^a,b,d^	185.30 ± 9.54 ^a,b,c^	<0.001 *
Ca (mg/dL)	9.84 ± 0.13	9.91 ± 0.12	9.79 ± 0.16	9.87 ± 0.09	0.61
P (mg/dL)	3.79 ± 0.11	3.82 ± 0.14	3.70 ± 0.19	3.81 ± 0.09	0.72
25 Hydroxycholecalciferol (nmol/L)	5.95 ± 1.08	110.00 ± 8.89 ^a,c,d^	4.44 ± 1.25 ^b,d^	75.49 ± 6.72 ^a,b,c^	<0.001 *
SMP-30 (pg/mL)	133.10 ± 19.20	170.50 ± 29.80 ^a,c,d^	58.50 ± 22.06 ^a,b^	66.30 ± 18.76 ^a,b^	<0.001 *

Statistical analysis was carried out using one-way analysis of variance (ANOVA) followed by Tukey’s post hoc analysis. ^a^ Significantly different from negative group at *p*-value < 0.05. ^b^ Significantly different from positive group at *p*-value < 0.05. ^c^ Significantly different from obese group at *p*-value < 0.05. ^d^ Significantly different from obese + vitamin D supplement group at *p*-value < 0.05. * *p*-value ≤ 0.05 is considered significant.

**Table 2 cells-10-00920-t002:** The gene expression of eNOS, SMP-30, and P53 in all studied animal groups.

	Negative Control *n* = 10	Positive Control (Vit D) *n* = 10	Obese Group *n* = 24	Obese + Vitamin D Supplement *n* = 13	ANOVA *p* Value
eNOS gene expression (2^−∆∆CT^)	1 ± 0.02	1.74 ± 0.21 ^a,c,d^	2.16 ± 0.25 ^a,b,d^	1.38 ± 0.19 ^a,b,c^	<0.001 *
SMP-30 gene expression (2^−∆∆CT^)	1 ± 0.03	1.14 ± 0.03 ^c^	0.55 ± 0.15 ^a,b,d^	1.15 ± 0.23 ^c^	<0.001 *
P53 gene expression (2^−∆∆CT^)	1 ± 0.02	1.74 ± 0.18 ^a,c,d^	2.26 ± 0.27 ^a,b,d^	1.48 ± 0.23 ^a,b,c^	<0.001 *

Statistical analysis was carried out using one-way analysis of variance (ANOVA) followed by Tukey’s post hoc analysis. ^a^ Significantly different from negative group at *p*-value < 0.05. ^b^ Significantly different from positive group at *p*-value < 0.05. ^c^ Significantly different from obese group at *p*-value < 0.05. ^d^ Significantly different from obese + vitamin D supplement group at *p*-value < 0.05. * *p*-value ≤ 0.05 is considered significant.

**Table 3 cells-10-00920-t003:** Correlation of hydroxycholecalciferol (nmol/L) level with anthropometric and biochemical parameters.

	All the Studied Population		Negative Control (Vit D) *n* = 10		Positive Control *n* = 10		Obese Group *n* = 24		Obese +Vitamin-D Supplement *n* = 13	
	*r*	*p*-value	*r*	*p*-value	*r*	*p*-value	*r*	*p*-value	*r*	*p*-value
Weight (gm)	−0.38	0.00 *	0.22	0.54	0.18	0.61	0.06	0.79	−0.07	0.82
Height (cm)	−0.12	0.87	−0.44	0.21	−0.53	0.11	−0.23	0.28	−0.31	0.31
BMI (gm/cm^2^)	−0.21	0.03 *	0.53	0.12	0.46	0.18	0.05	0.81	0.34	0.25
Insulin u/IU/mL)	−0.58	0.00 *	0.05	0.89	0.48	0.16	0.04	0.85	−0.29	0.34
Glucose (mg/dL)	−0.25	0.05 *	0.90	0.00	0.49	0.15	0.25	0.23	0.60	0.03
Homa-IR	−0.38	0.00 *	0.39	0.27	0.37	0.29	0.23	0.27	0.41	0.17
LDL-C (mg/dL)	−0.65	0.00 *	0.73	0.02 *	−0.14	0.61	−0.29	0.17	0.32	0.28
HDL-C (mg/dL)	0.91	0.00 *	−0.21	0.41	0.56	0.09	−0.02	0.91	0.21	0.49
Chol (mg/dL)	−0.80	0.00 *	0.15	0.67	0.32	0.37	−0.02	0.94	0.59	0.04 *
TG (mg/dL)	−0.51	0.00 *	−0.63	0.05 *	−0.02	0.95	0.27	0.20	0.16	0.60
hs-cTnt (pg/mL)	−0.27	0.04 *	−0.56	0.09 *	0.42	0.23	0.03	0.88	−0.10	0.73
CK (U/I)	−0.56	0.00 *	0.47	0.17	−0.02	0.94	0.11	0.61	−0.04	0.91
CK-MB (U/I)	−0.62	0.00 *	0.76	0.01 *	−0.21	0.55	0.26	0.23	−0.29	0.34
AST (U/I)	−0.82	0.00 *	−0.01	0.79	−0.61	0.05 *	−0.12	0.59	−0.09	0.76
ALT (U/I)	−0.75	0.00 *	−0.30	0.39	−0.13	0.72	−0.12	0.57	−0.14	0.66
eNOS (ng/mL)	−0.34	0.10	0.05	0.81	−0.00	0.91	0.04	0.84	−0.36	0.23
SMP-30 (pg/mL)	0.65	<0.00 *	0.29	0.42	0.16	0.65	−0.03	0.88	−0.17	0.58

Statistical analysis was carried out using Spearman’s correlation analysis. *r* = Spearman’s rank correlation coefficient, * *p*-value ≤ 0.05 is considered significant.

**Table 4 cells-10-00920-t004:** Correlation of SMP-30 (pg/mL) level with anthropometric and biochemical parameters.

	All the Studied Population		Negative Control (Vit D) *n* = 10		Positive Control *n* = 10		Obese Group *n* = 24		Obese + Vitamin-D Supplement *n* = 13	
	*r*	*p*-value	*r*	*p*-value	*r*	*p*-value	*r*	*p*-value	*r*	*p*-value
Weight (gm)	−0.80	<0.00 *	0.41	0.24	−0.22	0.55	0.01	0.65	0.61	0.03 *
Height (cm)	−0.19	0.16	−0.29	0.41	−0.33	0.35	−0.01	0.98	−0.34	0.26
BMI (gm/cm^2^)	−0.81	<0.00 *	0.31	0.38	0.10	0.78	−0.37	0.078	0.04	0.81
Insulin u/IU/mL)	−0.64	0.00	−0.13	0.72	0.24	0.51	0.19	0.39	−0.22	0.47
Glucose (mg/dL)	−0.24	0.04	0.06	0.86	−0.09	0.80	0.13	0.56	−0.00	0.99
Homa-IR	−0.31	0.00	0.11	0.77	−0.23	0.53	0.03	0.95	−0.33	0.35
LDL-C (mg/dL)	−0.73	0.00	−0.12	0.73	0.26	0.46	0.30	0.15	−0.04	0.90
HDL-C (mg/dL)	0.84	0.00	−0.67	0.04	0.50	0.14	0.22	0.31	−0.01	0.98
CHOL (mg/dL)	−0.82	0.00	0.12	0.75	−0.30	0.39	0.31	0.14	0.−01	0.98
TG (mg/dL)	−0.69	0.00	−0.36	0.31	0.50	0.14	0.13	0.53	0.11	0.72
hs-cTnt (pg/mL)	−0.41	0.00	−0.24	0.51	0.39	0.27	−0.11	0.36	−0.27	0.38
CK (U/I)	−0.64	0.00	0.41	0.24	0.66	0.04	−0.18	0.41	0.48	0.01
CK-MB (U/I)	−0.73	<0.00 *	−0.02	0.96	0.27	0.44	0.19	0.39	0.34	0.26
AST (U/I)	−0.85	<0.00 *	0.04	0.91	−0.39	0.26	−0.12	0.58	−0.40	0.17
ALT (U/I)	−0.83	<0.00 *	−0.49	0.15	−0.17	0.64	−0.16	0.45	−0.23	0.45
eNOS (ng/mL)	−0.47	<0.00 *	0.55	0.01	0.33	0.35	−0.26	0.23	0.47	0.10

Statistical analysis was carried out using Spearman’s correlation analysis. *r* = Spearman’s rank correlation coefficient, * *p*-value ≤ 0.05 is considered significant.

## Data Availability

Not applicable.

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
