# Peer review of "Cellular Senescence and Vitamin D Deficiency Play a Role in the Pathogenesis of Obesity-Associated Subclinical Atherosclerosis: Study of the Potential Protective Role of Vitamin D Supplementation"

_cells, 2021, doi:10.3390/cells10040920_

Round 1

Reviewer 1 Report

Please remove the phrase at page 3, line 3: "Female or non-albino rats...". At the beginning of the experiment there were only 60 male albino rats...so where were the female or non-albino rats or the 7 rats that failed to gain the expected weight or rats >200 gr in weight?

Please replace throughout the text the term "anthropometric" (it should be used only for measurements of human body) with "body measurements".

Discussion: Please cite: 1)De Pergola et al. Possible role of hyperinsulinemia and insulin resistance in lower vitamin D levels in overweight and obese patients. Biomed Res Int. 2013;2013:921348. 2) De Pergola et al. 25 Hydroxyvitamin D Levels are Negatively and Independently Associated with Fat Mass in a Cohort of Healthy Overweight and Obese Subjects. Endocr Metab Immune Disord Drug Targets. 2019;19(6):838-844.

Author Response

Dear Editor-in-chief,

Dear Reviewers,

I am very glad to receive your letter, thanks for your time, effort and support.

Please find below our point-by-point response for the manuscript cells-1146898. We have successfully replied to all of the reviewer´s comments and revised the manuscript according to their precious comments. All changes made in the revised version of our manuscript.

Yours;

Ayman Elsamanoudy

The corresponding author

Response to reviewers comments

Reviewer #1:

1

Please remove the phrase at page 3, line 3: "Female or non-albino rats...". At the beginning of the experiment there were only 60 male albino rats...so where were the female or non-albino rats or the 7 rats that failed to gain the expected weight or rats >200 gr in weight?

These are the exclusion criteria used for choosing rats used in the experiments. They include female, non-albino rats, rats weight more than 200gm. We start experiment with 67 rats with these exclusion criteria, 7 rats failed to gain weight and 3 rats died during the experiment period, so finally we have 57 rats included and continued the experiment.

This points are rewritten in the revised manuscript to be more clear.   

2

Please replace throughout the text the term "anthropometric" (it should be used only for measurements of human body) with "body measurements".

By revising the suitability of the term to be used for assessment of the body composition parameters in experimental animal, it is found that the term anthropometric can be used also for animal (rats as in the current study).

A lot of references presented it as anthropometric; for example:

  1. Mutiso SK, Rono DK, Bukachi F. Relationship between anthropometric measures and early electrocardiographic changes in obese rats. BMC Res Notes. 2014 Dec 18; 7: 931. doi: 10.1186/1756-0500-7-931. PMID: 25522784; PMCID: PMC4302031.
  2. Majeed KA, Ur Rehman H, Yousaf MS, Zaneb H, Rabbani I, Tahir SK, Rashid MA. Sub-chronic exposure to low concentration of dibutyl phthalate affects anthropometric parameters and markers of obesity in rats. Environ Sci Pollut Res Int. 2017 Nov;24(32):25462-25467. doi: 10.1007/s11356-017-9952-y.

3

Discussion: Please cite: 1)De Pergola et al. Possible role of hyperinsulinemia and insulin resistance in lower vitamin D levels in overweight and obese patients. Biomed Res Int. 2013;2013:921348. 2) De Pergola et al. 25 Hydroxyvitamin D Levels are Negatively and Independently Associated with Fat Mass in a Cohort of Healthy Overweight and Obese Subjects. Endocr Metab Immune Disord Drug Targets. 2019;19(6):838-844.

Cited in the revised manuscript.(reference number 21 & 39)

Reviewer 2 Report

In their study Bima et al. investigate the effect of a vitamin D substitution on the development of atherosclerosis in obese rats. The authors demonstrate a decreased expression of senescence marker protein 30 in obese rats, indicative of vascular senescence, which is abrogated by the substitution of vitamin D. They furthermore show an increase in eNOS and p53 gene expression in obese rats, which again is reduced in vitamin D supplemented animals, which indicates an amelioration of obesity induced endothelial activation.

Major comments:

  • Could the authors comment on the validity of their animal model. Even ApoE-/- mice on a high fat diet do not fully recapitulate human atherosclerosis while ApoE-/- rats on a high fat diet display even milder forms of atherosclerosis with little to no atherosclerosis reported in wild type rats on high fat diets (see for example Gao et al., J Biomed Res, 2017). Can the authors present evidence, that atherosclerosis is provoked in their model (e.g. fibrosis, macrophage invasion, intimal thickening).
  • Could the authors comment on the cell type specificity of their results? Since whole aortas have been used as samples it is hard to interpret without additional experiments (stainings?).
  • Is the increase in Troponin by a factor of three biologically relevant and what does it indicate? Sould it be interpreted as a direct damage to the myocardium upon a high fat diet or as a result of atherosclerosis mediated lack of myocardial perfusion? If the later is assumed, is there a histological indication of vascular dysfunction?
  • To display the data in tables rather than bar graphs (or a similar mode of presentation) is both visually unappealing as well as cumbersome and thus should be changed.
  • Are the changes in eNOS, SMP30 and p53 expression relevant i.e. is endothelial proliferation and NO-production altered and does this have an impact on physiological parameters (for example blood pressure).

Minor comments:

The manuscript should be carefully rewritten to allow for better reading comprehension. Groups should be given more descriptive names.

Author Response

Dear Editor-in-chief,

Dear Reviewers,

I am very glad to receive your letter, thanks for your time, effort and support.

Please find below our point-by-point response for the manuscript cells-1146898. We have successfully replied to all of the reviewer´s comments and revised the manuscript according to their precious comments. All changes made in the revised version of our manuscript.

Yours;

Ayman Elsamanoudy

The corresponding author

Response to reviewers comments

Reviewer #2:

1

Could the authors comment on the validity of their animal model. Even ApoE-/- mice on a high fat diet do not fully recapitulate human atherosclerosis while ApoE-/- rats on a high fat diet display even milder forms of atherosclerosis with little to no atherosclerosis reported in wild type rats on high fat diets (see for example Gao et al., J Biomed Res, 2017). Can the authors present evidence, that atherosclerosis is provoked in their model (e.g. fibrosis, macrophage invasion, intimal thickening).

Histological examination of the Aorta had been carried out, the intima/media thickness (IMT) was calculated and accordingly, the atherosclerosis was confirmed.

intima / media thickness (IMT) (μm) results were added to table 1

Results and Photos of histological examination:  page 8, 9 -lines 257-281

The following study proved the occurrence of atherosclerosis in high fat diet animal:

Subramani C, Rajakkannu A, Rathinam A, Gaidhani S, Raju I, Kartar Singh DV. Anti-atherosclerotic activity of root bark of Premna integrifolia Linn. in high fat diet induced atherosclerosis model rats. J Pharm Anal. 2017 Apr;7(2):123-128. doi: 10.1016/j.jpha.2016.12.002.

2

Could the authors comment on the cell type specificity of their results? Since whole aortas have been used as samples it is hard to interpret without additional experiments (stainings?).

Histological examination of the Aorta had been carried out. Results and Photos of histological examination: page 8, 9 -lines 257-283

3

Is the increase in Troponin by a factor of three biologically relevant and what does it indicate? Should it be interpreted as a direct damage to the myocardium upon a high fat diet or as a result of atherosclerosis mediated lack of myocardial perfusion? If the later is assumed, is there a histological indication of vascular dysfunction?

The increased level of cardiac troponin-T (hs-cTnT) in the current study in explained in Discussion sector in page 10 lines 353-360.

It is explained by the combined effect of obesity and subclinical atherosclerosis and its associated impaired circulation that could affect the cardiac perfusion, but still at the subclinical level.

As reported in cited references 43 and 44, obesity has an association with abnormal function and structure of myocardial muscle with an increased risk for the development of subclinical myocardial damage, as indicated by higher levels of cardiac troponin-T(hs-cTnT). So, obesity has been independently linked to elevated hs-cTnT levels, and the combination of obesity and elevated hs-cTnT is associated with increased risk of myocardial damage through the biochemically proved endothelial dysfunctions and subclinical atherosclerosis.

4

To display the data in tables rather than bar graphs (or a similar mode of presentation) is both visually unappealing as well as cumbersome and thus should be changed.

Data are presented in both tables and figures in the revised manuscript. Tables are used to present much data simultaneously that are present in our study. Two figures are added: Figure one to present gene expression in a more visually appealing way and Figure two for the histological examination that is must be presented in photos.

5

Are the changes in eNOS, SMP30 and p53 expression relevant i.e. is endothelial proliferation and NO-production altered and does this have an impact on physiological parameters (for example blood pressure).

eNOS is a marker of endothelial dysfunction and SMP30, is a marker of cellular senescence. P53 gene upregulation in adipose tissue promoted insulin resistance as reported by Chen et al., 2015 [reference no 34].

The mentioned parameters are relevant in the pathogenesis in obesity related endothelial dysfunction and its consequence subclinical atherosclerosis which could prove our hypothesis in which exaggerated cellular senescence is additional mechanism that participate in the pathogenesis of the obesity related comorbidities. 

Reviewer 3 Report

The paper was reviewed. The study verifies the possible role of vitamin D deficiency, and cellular senescence in the pathogenesis of obesity-related subclinical atherosclerosis. 

The paper was well written but still some conflicts necessary to be clarified.

Major:

  1. We believe that vitamin D supplementation could be beneficial to obesity-related atherosclerosis. However, the experiment still needs to explain the death of three rats during the obesity vitamin D deficiency groups. No death in the other groups.
  2. Is it possible to show the tissue pathological changes of the aorta- H.E. stain about atherosclerosis?
  3. Is it possible to show serum calcium and phosphate in the four groups?
  4. Is it possible to organize the evidence of the study to show a figure --a simple mechanism?

Author Response

Dear Editor-in-chief,

Dear Reviewers,

I am very glad to receive your letter, thanks for your time, effort and support.

Please find below our point-by-point response for the manuscript cells-1146898. We have successfully replied to all of the reviewer´s comments and revised the manuscript according to their precious comments. All changes made in the revised version of our manuscript.

Yours;

Ayman Elsamanoudy

The corresponding author

Response to reviewers comments

Reviewer 3 #  

Major

1

We believe that vitamin D supplementation could be beneficial to obesity-related atherosclerosis. However, the experiment still needs to explain the death of three rats during the obesity vitamin D deficiency groups. No death in the other groups.

First and second groups are the control groups: 1st group is a negative control groups where rats are kept on normal diet and the second group is positive control group where rats are kept on normal diet enriched with vit D supplementation. So, death of normal rats is not expected as there is no cause or risky of death.

Death of three rats in the experimental group that is suffered from obesity and vitamin D deficiency could be expected and explained by the hypothesis of our study that obesity is associated with occurrence of subclinical atherosclerosis that could be exaggerated to clinical atherosclerosis the died three rats. Moreover, this obesity group is associated with insulin resistance and dyslipidemia, another risky factors and possible death risk.

The 4th group where obese rats are kept on diet supplemented with vitamin D, proved the potential protective role of vitamin D hypothesized by our study and so, no death in this group provide more clinical evidence-base for this protective role.

2

Is it possible to show the tissue pathological changes of the aorta- H.E. stain about atherosclerosis?

Histological examination of the Aorta had been carried out. Results and Photos of histological examination: page 8, 9 -lines 257-283

3

Is it possible to show serum calcium and phosphate in the four groups?

 Serum calcium and phosphorus are added and presented in table 1 in the submitted revised manuscript.

4

Is it possible to organize the evidence of the study to show a figure --a simple mechanism?

 The evidence, conclusion and summary of the study is presented in a graphical abstract submitted with the revised manuscript.

Round 2

Reviewer 2 Report

The concerns about the validity of the rat atherosclerosis model have been addressed adequately. Nonetheless a disclaimer about the difficulty of transfering rodent atherosclerosis phenotypes to a human setting should be included in the text.

The incorporation of a graphical display of key findings is appreciated.

There are no further comments

Author Response

Response to reviewer 2:

The disclaimer about the difficulty of transferring rodent atherosclerosis phenotypes to a human setting is added and included in the revised resubmitted manuscript as a limitation of the current study at the end of discussion part with its supporting references.

Limitation of the study:

One of the main important limitation of the current study is about the difficulty of transferring rodent atherosclerosis phenotypes to a human setting as most rodent models of atherosclerosis do not completely mimic the pathophysiology of the atherosclerotic artery in human. It is reported that atherosclerotic plaques frequently could be manifested in the wall of the aorta as well as the large vessels and seldom to affect the coronary arteries as in human (Chen et al., 2016; Golforoush et al., 2020).  Consequently, the validity of the rat model of atherosclerosis used in current study is highly acceptable at level of large blood vessels but not at small blood vessels level that needs a different study model for addressing them. 

Reference

Chen WR, Chen YD, Tian F, Yang N, Cheng LQ, Hu SY, Wang J, Yang JJ, Wang SF, Gu XF. Effects of liraglutide on reperfusion injury in patients with ST-segment-elevation myocardial infarction. Circ Cardiovasc Imaging. 2016 doi: 10.1161/CIRCIMAGING.116.005146.

Golforoush P, Yellon DM, Davidson SM. Mouse models of atherosclerosis and their suitability for the study of myocardial infarction. Basic Res Cardiol. 2020 Nov 30;115(6):73. doi: 10.1007/s00395-020-00829-5. 

Reviewer 3 Report

My questions are well answered. However, I am not satisfied about the aorta pathology-H. E. staining-- not clear for readers.

1.loss of endothelium cells should be stained VCAM or von Willebrand factor.  

2.Sections from plastic-embedded tissues were stained with Lee’s methylene blue to see increasing thickness of the tunica intima and media and irregularity of the intimal surface

3. inflammatory cells could be stained by macrophage  such as ED-2 cells.

Author Response

Dear Editor-in-chief,

Dear Reviewers,

I am very glad to receive your letter, thanks for your time, effort and support.

Please find below our point-by-point response for the manuscript cells-1146898. We have successfully replied to all of the reviewer´s comments and revised the manuscript according to their precious comments. All changes made in the revised version of our manuscript.

Yours;

Ayman Elsamanoudy

The corresponding author

Reviewer comment

However, I am not satisfied about the aorta pathology-H. E. staining-- not clear for readers.

Reply to Reviewer 3

The current study aims to verify, a molecular level, link between vitamin D status, exaggerated cellular senescence and obesity, and to investigate the possible protective role of vitamin D supplementation against obesity-related subclinical atherosclerosis. The H & E staining and measuring the thickness of the intima and media for the aorta, both are carried out to validate the occurrence of subclinical atherosclerosis and consequently fulfill our aim. The detailed aortic pathology is out of our scope in current study, it could be the aim and scope of further study at level of cellular pathology and validation of presence of atherosclerosis at advanced level not just at early subclinical stage.